# Novel HSPG2 Gene Mutation Causing Schwartz–Jampel Syndrome in a Moroccan Family: A Literature Review

**DOI:** 10.3390/genes14091753

**Published:** 2023-09-02

**Authors:** Raffaella Brugnoni, Daria Marelli, Nicola Iacomino, Eleonora Canioni, Cristina Cappelletti, Lorenzo Maggi, Anna Ardissone

**Affiliations:** 1Neuroimmunology and Neuromuscular Diseases Unit, Department of Clinical Research and Development, Fondazione IRCCS Istituto Neurologico Carlo Besta, 20133 Milan, Italy; raffaella.brugnoni@istituto-besta.it (R.B.); nicola.iacomino@istituto-besta.it (N.I.); eleonora.canioni@istituto-besta.it (E.C.); cristina.cappelletti@istituto-besta.it (C.C.); 2Child Neurology Unit, Department of Pediatric Neuroscience, Fondazione IRCCS Istituto Neurologico Carlo Besta, 20133 Milan, Italy; daria.marelli@unimi.it (D.M.); anna.ardissone@istituto-besta.it (A.A.); 3Department of Biomedical and Clinical Sciences, Postgraduate School of Child Neuropsychiatry, University of Milan, 20157 Milan, Italy

**Keywords:** Schwartz–Jampel syndrome type 1, *HSPG2* gene, perlecan

## Abstract

Schwartz–Jampel syndrome type 1 (SJS1) is a rare autosomal recessive musculoskeletal disorder caused by various mutations in the *HSPG2* gene encoding the protein perlecan, a major component of basement membranes. We report a novel splice mutation *HSPG2*(NM_005529.7):c.3888 + 1G > A and a known point mutation *HSPG2*(NM_005529.7):c.8464G > A, leading to the skipping of exon 31 and 64 in mRNA, respectively, in a Moroccan child with clinical features suggestive of SJS1 and carrying two compound heterozygous mutations in the *HSPG2* gene detected by next-generation sequencing. Both parents harboured one mutation. Real-time and immunostaining analysis revealed down-regulation of the *HSPG2* gene and a mild reduction in the protein in the muscle, respectively. We reviewed all genetically characterized SJS1 cases reported in literature, confirming the clinical hallmarks and unspecific instrumental data in our case. The genotype–phenotype correlation is very challenging in SJS1. Therapy is mainly focused on symptom management and several drugs have been administered with different efficacy.Here, we report the second case with spontaneous improvement.

## 1. Introduction

Schwartz–Jampel syndrome type 1 (SJS1; OMIM #255800), also known as chondrodystrophic myotonia, is a rare autosomal recessive disorder characterized by permanent myotonic myopathy and skeletal dysplasia, which results in short stature, myotonia, chondrodysplasia, joint contractures, unusual pinnae, myopia and pigeon breast [1,2,3]. SJS1 is further divided into SJS1A, a milder phenotype, and SJS1B, a severe form with neonatal onset [4]. Myotonia, which is a major criterion for the diagnosis of SJS, may present at each age during the disease course. Affected children are diagnosed usually within the first years of life when myotonic signs and specific somatic features are noticed, among which are a distinctive facial appearance with a pointed mouth and narrowed eye fissures. Skeletal abnormalities include short stature and contractures of large joints [3].

The genetic basis had been reported in 2000 by Nicole and colleagues [5]. SJS is characterized by partial loss-of-function mutations distributed along the 97 exons of the *HSPG2* gene encoding perlecan, a ubiquitous heparan sulfate proteoglycan 2 secreted into basement membranes, which binds to growth factors and cell membrane receptors (OMIM*142461) [3,5,6]. Functional null mutations of the *HSPG2* gene are also associated to the Silverman–Handmaker type of dyssegmental dysplasia (DDSH; OMIM #224410), a less frequent and more severe autosomal recessive disorder. In this condition, the complete loss of perlecan function and lack of extracellular perlecan expression cause severe vertebral disc separation and thoracic dysplasia, leading to perinatal lethality [7].

Perlecan is a large protein (469 kD) composed of 4391 amino acids carrying three glycosaminoglycan chains at the N-terminus and five distinct domains. Domain I contains attachment sites for heperan sulfate chains and interacts with laminin, collagen and fibronectin; domain II is arranged into four cysteine-rich modules and binds low-density lipoprotein (LDL) receptor; domain III consists of an altering arrangement of three cysteine-free globular modules within a laminin epidermal growth factor-like motif; domain IV is made up of a tandem array of twenty-one immunoglobulin-like repeats; domain V is composed of a tandem arrangement of three laminin-type G and four EGF-like modules [3,8,9,10,11]. This ubiquitous protein binds to a wide range of molecules including growth factors and other multivalent extracellular matrix components, and has multiple functions, including growth factor signalling, cell adhesion, angiogenesis, basement membrane and cartilage maintenance and acetylcholinesterase anchoring at the neuromuscular junction, and plays an important role in maintaining cartilaginous tissue integrity and regulating muscle excitability [12,13].

In this study, we report a Moroccan child with clinical features of SJS1 carrying two compound heterozygous mutations in the *HSPG2* gene: one novel *HSPG2*(NM_005529.7):c.3888 + 1G > A, and one known *HSPG2*(NM_005529.7):c.8464G > A. In addition, we reviewed all cases with confirmed genetic diagnosis reported in literature.

## 2. Case Report

### 2.1. Patient

The proband is the second child of consanguineous Moroccan parents (third-grade cousins). His family history was unremarkable. He was born at term after an uneventful pregnancy. He presented mild motor delay, being able to walk alone at 20–24 months of age, whereas language and cognitive development were determined to be normal. Since early years, he suffered from stiffness and pain in hand muscles.

He was first evaluated at 18 months of age, presenting blepharophimosis, micrognathia and low-set ears; neurological examination showed mild motor delay, and spontaneous motility was characterized by stiffness due to myotonia. The creatine kinacse (CK) level was elevated (750 U/L), and electromyograph (EMG) revealed myotonic discharges with high-frequency repetitive activity. First, myotonic dystrophy type I and Becker myotonia were ruled out via genetic investigation and a muscle biopsy revealed mild changes with cores.

The symptoms worsened over time: at 5–6 years of age, he presented stiffness and myalgias in both upper and lower arms and facial muscles, impairing walking ability and chewing.

He was assessed at 6 years of age. On physical examination, facial muscle stiffness appeared to have worsened (Figure 1A): he showed pursed lips, almost completed blepharophimosis, and speaking was very difficult due to oromuscolar involvement. Global hypertrophy was evident and motility was completed but clumsy and limited by stiffness. The Gower sign was positive. Myotonia was not observed, but difficulty in muscle release after maximum contraction of the quadriceps was disclosed. He also presented skeletal deformities such as coxa valga and limitations of joint range of motion. The CK was 800 U/L, while other blood tests, including potassium and sodium, were normal. EMG showed myopathic abnormalities associated with repetitive activity almost subcontinua, also with myotonic aspects. SJS was suspected based on clinical findings.

Carbamazepine therapy was not effective. At the following routine evaluation, the patient showed a clinical spontaneous favourable evolution with functional improvement. At the last visit, at the age of 10 years, facial appearance was stable (Figure 1B), but he presented a further functional slight improvement in chewing function and speaking. He underwent surgery for blepharophimosis, without significant improvement. The gait was still characterized by stiffness, but functional limitations were not reported yet. On examination, we observed persistence in the difficulty of muscle release after maximum contraction limited to the hamstrings, and a global reduction in muscle hypertrophy. Treatment with mexiletine was proposed, but the family refused due to the spontaneous improvement.

### 2.2. Genetic Analysis for the HSPG2 Gene

The genomic DNA of the patient and their parents was extracted from peripheral blood, as previously reported [14]. A specific next generation sequencing (NGS) panel that included the whole *HSPG2* gene was designed using Sure Design (https://earray.chem.agilent.com/suredesign/; accessed on 16 July 2023) (Agilent Technologies, Santa Clara, CA, USA). DNA libraries of the patient were prepared as described elsewhere [14]. To identify pathogenic variants and exclude variants with an allele frequency of more than 1% (MAF > 0.01), public databases (i.e., dbSNP, 1000 Genome Project, ExAC, ClinVar and HGMD) were used.

The NGS panel allowed us to identify two heterozygous mutations in the proband (Figure 2). For one allele, patient II1 presented a novel splice mutation, i.e., *HSPG2*(NM_005529.7):c.3888 + 1G > A, that affected the more conserved GT dinucleotide sequence of 5′ splice sites of intron 31 of the gene. According to the American College of Medical Genetics (ACMG) guidelines, this variant was classified as “likely pathogenic” [15]. For the second allele, a published G > A transition was found in the last nucleotide, *HSPG2*(NM_005529.7):c.8464G > A of exon 64, classified as “uncertain significance” following the ACMG guidelines, and resulted in the skipping of exon 64 already described by Arikawa-Hirasawa and colleagues [3]. The two variants discovered via NGS were validated by Sanger sequencing using the following primers (designed using Primer3): forward—GTGGTGAGCAGAGTGGAGG, and reverse—AGATGCTGCCTGATTTCCCC for exon 31; and forward—GCACCTGACTCTGTTCCTCT and reverse—CTGGTCTCTGTCCCTTCCTC for exon 64 (Figure 2B). The results were analyzed using SeqScape v.3 software (Thermo Fisher, Foster City, CA, USA) and compared with the GeneBank reference sequence of the *HSPG2* gene (NM_005529.5). Sanger sequencing for the proband’s parents confirmed that the father, I1, was a carrier of the variant in exon 64 and the mother, I2, was a carrier of the variant in intron 31 of the *HSPG2* gene (Figure 2B).

The localization of the founded mutations was in domains III and IV of the perlecan protein (Figure 2C). The novel splice site mutation *HSPG2*(NM_005529.7):c.3888 + 1G > A was conserved among a diverse range of species (Figure 2D).

### 2.3. mRNA Analyses

Total RNA was extracted from a frozen muscle biopsy using TRIzol^®^ Reagent, and then reverse-transcribed using the SuperScript^®^ VILO cDNA Synthesis kit (both from Invitrogen, Waltham, MA, USA, Thermo Fisher Scientific, Waltham, MA, USA), following the manufacturer’s instructions.

We analyzed the novel *HSPG2*(NM_005529.7):c.3888 + 1G > A mutation by amplifying the complementary DNA of the *HSPG2* mRNA via PCR using a specific primer set within the exons flanking the exon 31 (forward in exon 29, TGCCCACACTTGTTTTCTGG; reverse in exon 32, GGTGATGCCCATACAGAAGC). The 35 PCR cycles consisted of a denaturation step at 96 °C for 30 s, an annealing step at a 60 °C temperature for 30 s, and an elongation step at 72 °C for 1 min. The PCR product loaded on a 2% agarose gel revealed a second band of a lower height (238 bp) in the patient than the control sample (333 bp), corresponding to the excision of the exon 31 (long 95 bp) (Figure 3A). The sequencing of the PCR product allowed us to confirm the heterozygous excision of the entire exon 31 (Figure 3B).

### 2.4. Real-Time PCR Analysis of HSPG2

To determine whether the level of *HSPG2* mRNA was reduced, we quantitatively analysed *HSPG2* mRNA from the muscle of the patient and six healthy controls, performing a quantitative real-time PCR employing a ViiA™ 7 Real-Time PCR System (Applied Biosystems, Waltham, MA, USA, Thermo Fisher Scientific). We used a pre-designed functionally tested TaqMan™ Gene Expression Assay for the *HSPG2* gene and, as internal control, the *GAPDH* gene (Applied Biosystems). Real-time PCR reactions were carried out in triplicate for the patient and in duplicate for the healthy control samples. Transcriptional levels were calculated as a relative expression using the formula 2^−ΔΔCt^ by normalizing the patient’s mean ΔCt value with the control mean ΔCt values.

We found that the transcriptional level of *HSPG2* mRNA from the muscle of the patient is lower than the control (0.37 vs. 0.86 ± 0.35), indicating a decrease in the perlecan-mRNA of the patient caused by the presence of the two described mutations of the *HSPG2* gene in our patient (Figure 4A).

### 2.5. Immunostaining Analysis

Next, we examined the expression of perlecan protein in muscle tissue sections from the patient and control via double immunofluorescence using a specific domain V anti-perlecan antibody and, as a positive control, anti-fibronectin antibody (Figure 4B). The immunostaining protocol was adapted from [16]. Briefly, cryosections were fixed in 4% paraformaldehyde, permeabilized in 0.1% Triton X-100 and then blocked in 5% bovine serum albumin in PBS. Mouse monoclonal to perlecan (anti-heparan sulfate proteoglycan antibody, 1:100 dilution, abcam, Cambridge, UK) and rabbit polyclonal to fibronectin (anti-fibronectin antibody; 1:50 dilution, abcam) primary antibodies were incubated overnight at 4 °C. The slides were then incubated with secondary antibodies (goat anti-mouse Alexa Fluor™ 488, goat anti-rabbit Alexa Fluor™ 546, both from Invitrogen) for 1 h at room temperature, counterstained with DAPI (Invitrogen) and afterward mounted with FluorSave™ Reagent (Merck, Darmstadt, Germany). The images were acquired with a Nikon Eclipse TE2000-E confocal laser scanning microscope (Nikon, Minato City, Tokyo, Japan) and analyzed with ImageJ software (version 1.52p, National Institutes of Health, Bethesda, MD, USA).

Immunostaining analysis revealed that in the control tissue, perlecan was localized in the basal lamina of the muscle fibres, while weaker staining was observed in the muscle of the patient (Figure 4B). Fibronectin localization with the ECM appeared to be normal in both the patient and control tissues. Further, we quantitatively analyzed perlecan protein by measuring the fluorescent area of five randomly selected fields, both in the control and patient samples, using ImageJ software. Our quantitative analysis of perlecan fluorescence, normalized by the fibronectin fluorescent area, confirmed a statistically significant minor amount of perlecan protein in the patient muscle fibre (*p* < 0.01, Figure 4C).

## 3. Discussion

SJS1 is a rare autosomal recessive progressive disorder, caused by an abnormality in the perlecan gene, with myotonic symptoms and cartilage abnormalities [3,10,11].

We reviewed genetically defined reported cases (see Table 1). There are other anecdotal cases with a clinical diagnosis, without molecular confirmation, hence not included in this review. More than 40 mutations distributed along the whole *HSPG2* gene associated with SJS1 causing a loss-of-function of the perlecan protein (Table 1 and Figure 5) have been published to date [17,18,19,20,21,22,23]. Unfortunately, clinical features have not been described in detail in all reports, and treatment or outcome data have been reported only in some cases.

The clinical data reported in the literature showed myotonic signs in all patients and typical facial features due to myotonia and skeletal dysplasia, in accordance with the definition of the syndrome. Other symptoms or signs were not reported, and phenotypic variability was not disclosed. The typical phenotype was confirmed in our patient. Our case, together with data from the literature, suggests that the clinical picture is pathognomonic of this syndrome, and instrumental examinations such as EMG and biopsy do not provide any additional clues useful for the diagnosis of SJS1. Hence, SJS1 diagnosis mainly relies on clinical data, then needing molecular confirmation.

Therapy mainly focuses on symptom management and primarily aims to reduce myotonia. Therapeutic approaches vary considerably among the reports, making it challenging to establish recommendations (Table 1). However, anticonvulsivants, such as carbamazepine or phenytoin, blocking sodium channels with stabilization of the membrane of excitable tissues, represent the most used drugs, followed by antiarrhythmic therapies such as mexiletine, procainamide and quinidine. To reduce localized myotonia, especially for mask-like facies, some studies reported the injection of botulinum toxin or surgical approaches. However, as mentioned before, it is very hard to draw any conclusion from the literature due to the relative lack of data on the efficacy of different approaches and considering that most of the studies include small case series.

In our case, we decided to start treatment for reducing myotonia with carbamazepine, however, without any benefit. At the last visit, we proposed a treatment with mexiletine after a cardiac evaluation, but the family decided not to start any further pharmacological treatment for myotonia due to the slight spontaneous functional improvement.

Due to extra neurological sequelae, the proband undergoes periodic cardiological evaluations, comprising an electrocardiogram Holter and echocardiogram, ultrasound of the abdomen, orthopaedic evaluation and an oculist follow-up in order to monitor blepharophimosis.

We have observed a potential spontaneous improvement in muscle symptoms in our patients. To our knowledge, this is only the second case where such a symptomatic and functional improvement has been reported. The other case was reported by Bauchè [20]: the patient presented early onset in the first year of age and disclosed a typical clinical picture in childhood; phenytoin was administrated in infancy without efficacy. In the following years, he showed spontaneous facial and general stiffness improvement, but details are not available. The patient harboured compound mutations in the I and III domain, different from our cases.

Domains III and IV of the perlecan protein have the highest number of mutations identified so far in the *HSPG2* gene, with 14 and 16 mutations, respectively (Figure 5), though domain V had *HSPG2*(NM_005529.7):c.11208-7G > A as the most frequent mutation identified, carried in five patients of different ethnicities (four from Saudi Arabia and one from Tunisia) [10]. The other two more frequent mutations in the population were *HSPG2*(NM_005529.7):c.8464 + 4A > G and *HSPG2*(NM_005529.7):c.4741-10T > G, both identified in three Tunisian patients (Table 1) [5,10]. Among the 44 different mutations identified along the *HSPG2* gene, the most common were 16 splice, 11 deletions and 10 missense mutations (Table 1).

In the present study, we identified the compound heterozygous state of the *HSPG2* gene in a Moroccan child with clinical features of SJS1: one novel splice site variant *HSPG2*(NM_005529.7):c.3888 + 1G > A and one reported *HSPG2*(NM_005529.7):c.8464G > A pathogenic mutation [6], inherited from his asymptomatic parents (Figure 2). Both mutations result in whole-exon skipping, exon 31 for c.3888 + 1G > A (Figure 3) and exon 64 for c.8464G > A [6], resulting in two aberrant splicing transcripts predicted to produce shorter-than-normal perlecan molecules on both alleles.

Another *HSPG2*(NM_005529.7):c.8464 + 4A > G splice mutation in other SJS1 patients [5,10] was also described in the c.8464 position of the perlecan, confirming that this is a likely hot spot region for mutations in SJS1 patients ([5,6,10] our study).

Considering the position of two mutations on domains III and IV of the protein (Figure 2C), they may cause a decrease in the homologous-to-the-laminin short-chain N-terminal region and the immunoglobulin-like structural sites of neural cell adhesion molecule, respectively [3]. Moreover, the loss of the C-terminal domain may be responsible for the early onset of skeletal abnormalities in our patient, as previously reported [6], because the immunostaining of perlecan in muscle tissues of the patient compared to the control is defective in domain V (Figure 4), whereas this region plays a role in matrix–cell interactions through its binding to a chondrocyte receptor, providing stable formation of the cartilage matrix structure.

## 4. Conclusions

In conclusion, we presented the first case of a family Moroccan origin carrying two mutations in the *HSPG2* gene, inducing SJS1: one novel *HSPG2*(NM_005529.7):c.3888 + 1G > A and one described *HSPG2*(NM_005529.7):c.8464G > A, both causing adjacent exon skipping and a loss of function of perlecan. These two mutations were localised in domains III and IV of the perlecan protein that, according to our literature review, have the highest number of mutations identified so far. They can be considered as a hot-spot region of the *HSPG2* gene associated with the SJS1 phenotype. Most mutations are private and several pathogenic mutations, including splice site variants resulting in whole-exon skipping or creating premature termination codons, were reported in this gene in association with SJS1. In Table 1, we described the cases’ ethnicities to investigate a possible founder effect. However, even in populations where several cases have been reported (i.e., Tunisia or Turkey), different mutations were described, often involving different domains, even within the same ethnicity; thus, we concluded that there is no evidence for a founder effect.

A literature review of the published genetically confirmed cases, and even our patient, confirmed that the clinical picture—and not instrumental data—is fundamental for diagnosis and to guide molecular analysis. Furthermore, pharmacological therapies aiming to reduce symptoms of myotonia are available; however, their efficacy needs further investigation.

## Figures and Tables

**Figure 1 genes-14-01753-f001:**
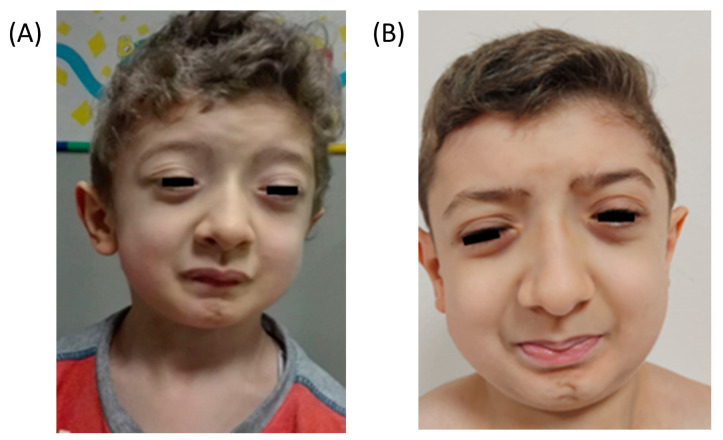
Clinical features of the patient at 6 years old (**A**) and 10 years old (**B**): mask-like facies with blepharophimosis, pursed lips, micrognathia and low-set ears.

**Figure 2 genes-14-01753-f002:**
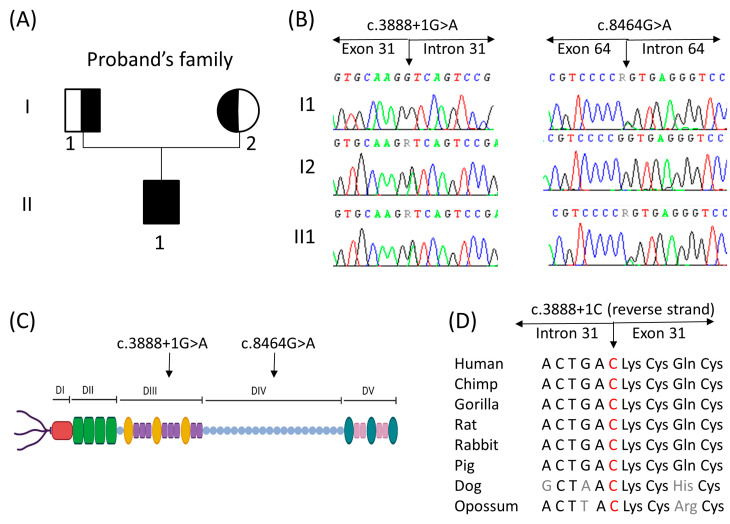
(**A**) Pedigree of the family. Distant consanguinity of the proband’s parents (third cousins) is reported. The proband (II1) was a compound heterozygote for mutations *HSPG2*(NM_005529.7):c.3888 + 1G > A and *HSPG2*(NM_005529.7):c.8464G > A in the *HSPG2* gene. The asymptomatic parents (I1 and I2) were carriers for one mutation of the offspring. (**B**,**C**) Electropherogram of the two identified mutations in all family members tested and their localization on a schematic representation of perlecan. (**C**) Figure was created using Biorender.com (Biorender, Toronto, ON, Canada; accessed on 1 September 2023). (**D**) The amino acid alignment around the novel splice site mutation c.3888 + 1G > A was evolutionally conserved across a variety of different species.

**Figure 3 genes-14-01753-f003:**
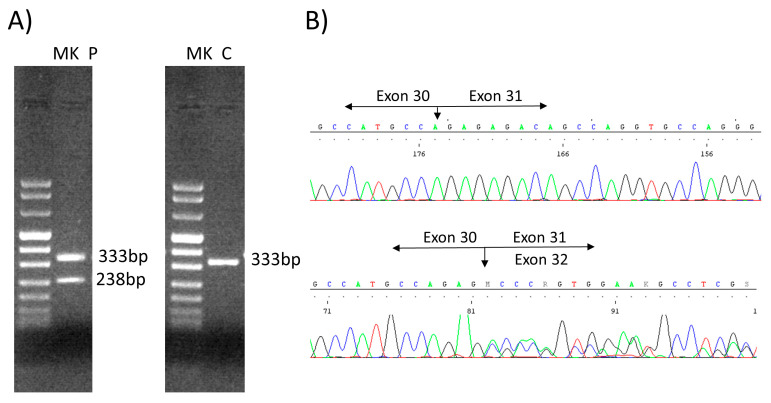
(**A**) Agarose (2%) gel. The amplification of the proband’s cDNA with primers forward exon 29 and reverse exon 32, revealed two bands (sample P: 333 bp and 238 bp), unlike the control (sample C: 333 bp). (**B**) The sequencing of the amplification products from the patient compared with the control (**up**) revealed the excision of the entire exon 31 (95 bp) from the *HSPG2*-cDNA (**down**). P, patient; C, control; MK, marker VIII (Bio-Rad, Hercules, CA, USA).

**Figure 4 genes-14-01753-f004:**
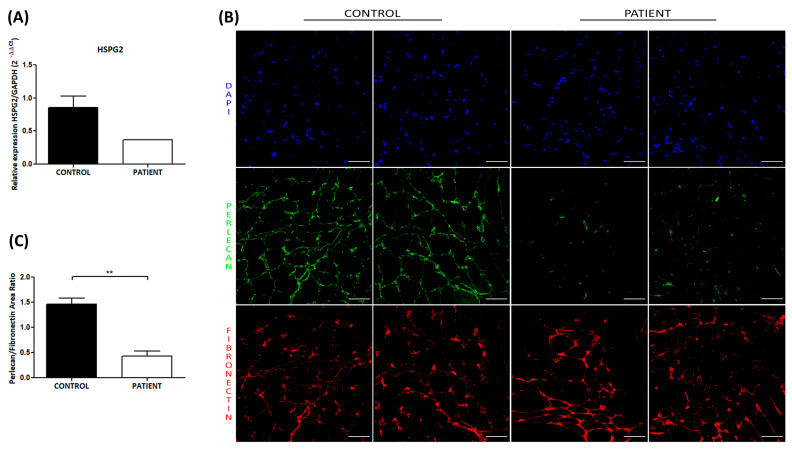
Decreased expression of perlecan in the muscle tissue of the patient was observed via real-time PCR and immunofluorescence analysis. (**A**) Real-time PCR analysis of *HSPG2* mRNA in muscle tissue from the patient and healthy controls showed a reduction in *HSPG2* transcriptional levels in the patient tissue. Transcriptional levels of *HSPG2* are expressed as the 2^−ΔΔCt^ value ± SEM. (**B**) Representative immunostaining images of perlecan in the muscle tissues from patient and control subject. Double immunostaining showed domain V (green) of perlecan and fibronectin (red) within the extracellular matrix. The staining revealed that the domain-V-stained the extracellular matrix at significantly reduced levels compared to the control, whereas fibronectin staining is strong in both tissues. Blue staining shows DAPI-positive nuclei. Scale bar: 50 µm. (**C**) Quantification of perlecan immunofluorescence in both the patient and control subject reported as the ratio ± SEM of perlecan fluorescent area normalized by the fibronectin fluorescent area. The quantitative analyses showed a significant reduction in perlecan protein in patient muscle tissue compared to the control tissue. The statistical analyses were performed through the Mann–Whitney *t*-test to compare non-parametric data, using GraphPad Prism version 5.0 (GraphPad Software, San Diego, CA, USA). ** *p* < 0.01.

**Figure 5 genes-14-01753-f005:**
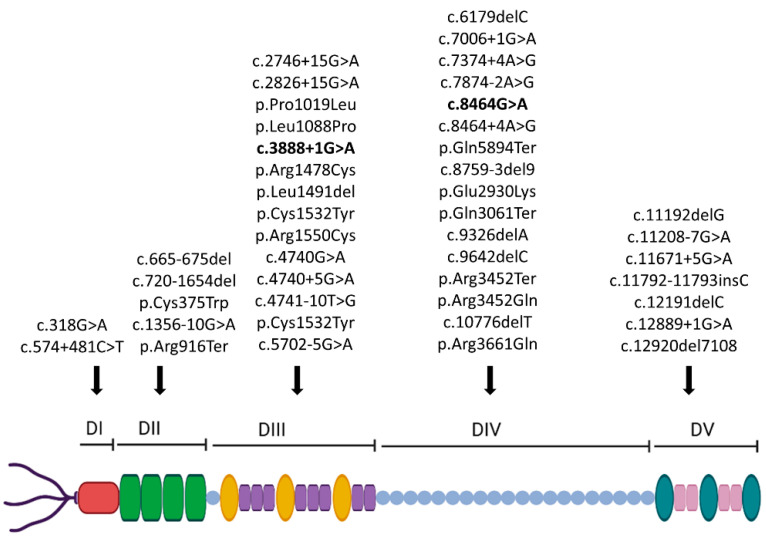
Localization of all the variants summarized in this review on a schematic representation of perlecan (the two mutations identified in our family are in bold).

**Table 1 genes-14-01753-t001:** Previously reported missense mutations in the *HSPG2* gene in SJS and the two mutations found in our Moroccan patient (in bold). (*), mutations present in homozygous form; NR, not reported; m, month; y, year; Cbz, carbamazepine; λ, genetical aspects firstly described in Nicole et al., clinical aspects reported in Stum et al. Mx, mexiletine; Ph, phenytoin; NI, second mutant allele was not identified; UdE, undetermined effect; #, Only short stature has been reported, but no other specific skeletal abnormalities.

Reference	N. Patient	Ethnicity	Sex	Age at onset	Mask-Like Facies	General Myotonic Signs	Skeletal Abnormalities	High Values of CPK	Muscle Biopsy	EMG	Medical Therapy Improvement	Spontaneous Improvement	cDNA Mutation *HSPG2*(NM_005529.7)	Effect and Domain on Protein
Nicole 2000 [5], Stum 2006 [10] λ	2	Tunisia	NR	pt1: 2 y pt2: 3 y	Yes	Yes	Yes	NR	NR	doubtful neurogenic pattern	NR	NR	c.4740G > A *	UdE/III
Nicole 2000 [5], Stum 2006 [10] λ	2	Turkey	F	1 y	Yes	Yes	Yes	NR	NR	doubtful neurogenic pattern	NR	NR	c.4595G > A *	p.Cys1532Tyr/III
Nicole 2000 [5], Stum 2006 [10] λ	3	Tunisia	NR	2.5 y	Yes	Yes	Yes	NR	NR	doubtful neurogenic pattern	NR	NR	c.8464 + 4A > G *	p.Thr2773ProfsTer25/IV
Arikawa-Hirasawa 2002 [6]	1	NR	M	3 y	Yes	Yes	Yes	Yes	NR	myotonic signs	NR	NR	1: c.7374 + 4A > G 2: exon fusion 60–61	1: skip exon 56/IV 2: UdE/IV
Arikawa-Hirasawa 2002 [6]	1	NR	M	4 m	Yes	Yes	Yes	NR	Necrotic and regenerating fibres	myotonic signs	NR	NR	1: c.8464G > A 2: c.8759-3del9	1: skip exon 64/IV 2: retrain intron 66 or skip exon 67/IV
Arikawa-Hirasawa 2002 [6]	1	NR	M	3 m	Yes	NR	Yes	NR	NR	myotonic signs	NR	NR	c.12920del7108 *	retain intron 95 or 94–95/V
Stum 2006 [10]	1	Turkey	NR	6 m	Yes	Yes	Yes	NR	NR	doubtful neurogenic pattern	NR	NR	c.574 + 481C > T *	p.Val192AlafsTer25/I
Stum 2006 [10]	1	India	NR	2 y	Yes	Yes	Yes	NR	NR	NR	NR	NR	1: c.665-675del 2: NI	1: p.Arg222GlnfsTer5/II 2:NI
Stum 2006 [10]	1	Ireland	NR	2 y	Yes	Yes	Yes	NR	NR	NR	NR	NR	1: c.3055C > T 2: c.10355G > A	1: p.Pro1019Leu/III 2: p.Arg3452Gln/IV
Stum 2006 [10]	1	Mauritius	NR	NR	Yes	Yes	Yes	No	NR	NR	NR	NR	c.4432C > T *	p.Arg1478Cys/III
Stum 2006 [10]	1	Turkey	NR	6 m	Yes	Yes	Yes	NR	NR	doubtful neurogenic pattern	NR	NR	c.720-1654del *	UdE/II
Stum 2006 [10]	3	Tunisia	NR	2.5 y	Yes	Yes	Yes	NR	NR	doubtful neurogenic pattern	NR	NR	c.4741-10T > G *	UdE/III
Stum 2006 [10]	1	Turkey	NR	2 y	Yes	Yes	Yes	NR	NR	myotonic signs	NR	NR	c.4648C > T *	p.Arg1550Cys/III
Stum 2006 [10]	1	France	NR	2 y	Yes	Yes	Yes	NR	NR	myotonic signs	NR	NR	1: c.6179delC 2: NI	1: p.Pro2060LeufsTer3/IV 2:NI
Stum 2006 [10]	1	Albania	NR	birth	Yes	Yes	Yes	NR	NR	myotonic signs	NR	NR	1: c.8680C > T 2: c.10982G > A	1: p.Gln2894Ter/IV 2: p.Arg3661Gln/IV
Stum 2006 [10]	2	South Africa	NR	birth	Yes	Yes	Yes	NR	NR	myotonic signs	NR	NR	c.7006 + 1G > A *	UdE/IV
Stum 2006 [10]	2	Brazil	NR	9 m	Yes	Yes	Yes	NR	NR	Pt 1: doubtful neurogenic pattern Pt 2: myotonic signs	NR	NR	c.9326delA *	p.His3109ProfsTer16/IV
Stum 2006 [10]	1	Belgium	NR	2 y	Yes	Yes	Yes	NR	NR	doubtful neurogenic pattern	NR	NR	1: c.10354C > T 2: c.4432C > T	1: p.Arg3452Ter/IV 2: p.Arg1478Cys/III
Stum 2006 [10]	1	France	NR	4 y	Yes	Yes	Yes	NR	NR	myotonic signs	NR	NR	1: c.9642delC 2: c.7874-2A > G	1: p.Gln3215Lysfs Ter7/IV 2: p.His2624_Val2625ins39/IV
Stum 2006 [10]	1	France	NR	1 y	Yes	Yes	Yes	NR	NR	myotonic signs	NR	NR	1: c.10355G > A 2: c.4473_4475del	1: p.Arg3452Gln/IV 2: p.Leu1491del/III
Stum 2006 [10]	1	The Netherlands	NR	18 m	YES	Yes	Yes	NR	NR	doubtful neurogenic pattern	NR	NR	1: c.10982G > A 2: c.11192delG	1: p.Arg3661Gln/IV 2: p.Gly373GlufsTer30/V
Stum 2006 [10]	1	Tunisia	NR	1 y	Yes	Yes	Yes	NR	NR	myotonic signs	NR	NR	c.11208-7G > A *	UdE/V
Stum 2006 [10]	4	Saudi Arabia	NR	Pt1: 1 m Pt2: 6 m Pt3: 9 m Pt4: 2 y	Yes	Yes	Yes	NR	NR	myotonic signs	NR	NR	c.11208-7G > A *	UdE/V
Stum 2006 [10]	2	Indian	NR	8 m	Yes	Yes	pt1: Yes pt2: #	NR	NR	doubtful neurogenic pattern	NR	NR	1: c.11792-11793insC 2: NI	1: p.Leu3932AlafsTer32/V 2:NI
Stum 2006 [10]	1	Turkey	NR	3 y	Yes	Yes	No	NR	NR	myotonic signs	NR	NR	1: c.12191delC 2: c.4432C > T	1: p.Pro4065ArgfsTer5/V 2: p.Arg1478Cys/III
Stum 2006 [10]	1	Turkey	NR	3 m	Yes	Yes	Yes	NR	NR	doubtful neurogenic pattern	NR	NR	c.12899 + 1G > A *	UdE/V
Bauche 2013 [20]	1	NR	M	first 1 y	Yes	Yes	Yes	No	immunostaining: reduced perlecan in the extra-synaptic BM + fiber type I predominance	Myotonic signs	Ph/No	Yes	1: c.2746C > T 2: c.2826 + 15G > A	1: p.Arg916Ter/II 2: UdE/III
Iwata 2015 [2]	1	Japanese	M	2 y 3 m	Yes	Yes	Yes	No	fiber size variation, fibre degeneration, fiber type I predominance with grouping	myotonic signs	Cbz and Mx/Yes; Ph (s.e.)	NR	1: c.3263T > C 2: c.9181C > T	1: p.Leu1088Pro/III 2: p.Gln3061Ter/IV
Dai 2015 [22]	1	Chinese	F	8 m	Yes	Yes	Yes	Yes	NR	myotonic signs	Cbz/No	NO	1: c.5702-5G > A 2: c.10776delT	1: UdE/IV 2: p.Ala3592fsTer6/IV
Bhowmik 2016 [21]	1	Indian	M	2 y	Yes	Yes	Yes	Yes	NR	myotonic signs	Cbz (s.e.); Mx/Yes	NR	c.4740 + 5G > A *	p.Asp1543_Ser1580del/III
Padmanabha 2018 [19]	1	Indian	M	2 y	Yes	Yes	Yes	Yes	NR	myotonic signs	NR	NR	1: c.1356-10G > A 2: c.10355G > A	1: UdE/II 2: p.Arg3452Gln/IV
Yan 2018 [1]	1	Chinese	F	2 y	Yes	Yes	Yes	NR	NR	NR	NR	NR	1: c.8788G > A 2: c.11671 + 5G > A	1: p.Glu2930Lys/IV 2: UdE/V
Gurbuz 2019 [17]	1	Turkey	F	2.5 y	Yes	Yes	Yes	NR	NR	myotonic signs	Cbz/Yes	No	c.318A > G *	UdE/I
Lin 2021 [18]	2	Taiwan	pt1: M Pt2: F	Pt1: Childhood Pt2: NR	Yes	Yes	pt1: Yes pt2: NR	Yes	NR	pt1: myotonic signs pt2: NR	NR	NR	c.1125C > G *	p.Cys375Trp/II
**This study**	**1**	**Marocco**	**M**	**18 m**	**Yes**	**Yes**	**Yes**	**Yes**	**unspecific muscle suffering with specific aspects such as “core”**	**myotonic signs**	**Cbz/No**	**Yes**	**1: c.3888 + 1G > A** **2: c.8464G > A**	**1: skip exon 31/III** **2: skip exon 64/IV**

## Data Availability

The data presented in this study are openly available in a repository (https://doi.org/10.5281/zenodo.8019946, accessed on 16 July 2023) upon reasonable request.

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
