# Peer review of "Novel HSPG2 Gene Mutation Causing Schwartz–Jampel Syndrome in a Moroccan Family: A Literature Review"

_genes, 2023, doi:10.3390/genes14091753_

Round 1

Reviewer 1 Report

In the present manuscript, it suggests that the two detected compound heterozygous mutations in the HSPG2 gene may cause Schwartz-Jampel Syndrome type 1 (SJS1) and reports in a Moroccan child with clinical features. This paper is well-described. I recommend that this paper be accepted after minor revision.

1. In Real-time section, were the result (0.37 vs 1.23) obtained from one well each but not replicates? If qPCR were performed with replicates, it is better to add standard deviation (error bars). Are the values (0.37 and 1.23) the value that HSPG2 divided by GAPDH? If so, the expression of HSPG2 might be too high.  

2. It would be better to indicate the sequences and positions of primers used in quantitative real-time PCR.

3. It would be better to discuss the possibility that HSPG2 mRNA in patient was lower than in control. Is the expression of HSPG2 in the father with the novel splice site mutation also lower due to NMD?

4. In Figure 4, DAPI staining in control were faint. It is better to change to bright pictures.

Author Response

Comments and Suggestions for Authors:

In the present manuscript, it suggests that the two detected compound heterozygous mutations in the HSPG2 gene may cause Schwartz-Jampel Syndrome type 1 (SJS1) and reports in a Moroccan child with clinical features. This paper is well-described. I recommend that this paper be accepted after minor revision.

  1. In Real-time section, were the result (0.37 vs 1.23) obtained from one well each but not replicates? If qPCR were performed with replicates, it is better to add standard deviation (error bars). Are the values (0.37 and 1.23) the value that HSPG2 divided by GAPDH? If so, the expression of HSPG2 might be too high.  

Our results were shown as mean value of 2-ΔΔCt, obtained from a qPCR with replicates for the patient and 6 healthy controls. We have now included the standard error values as suggested. We calculated the 2-ΔΔCt for each replicate using Ct value of GAPDH as normalizer.

  1. It would be better to indicate the sequences and positions of primers used in quantitative real-time PCR.

We used pre-designed functionally tested TaqMan™ Gene Expression Assay, which includes target primers and a sequence-specific probe.

  1. It would be better to discuss the possibility that HSPG2 mRNA in patient was lower than in control. Is the expression of HSPG2 in the father with the novel splice site mutation also lower due to NMD?

We expect that the HSPG2 mRNA in the patient is lower than in the control, but we have not performed a WB to verify it, we preferred to directly verify the reduction of expression of the perlecan protein with an immunofluorescence experiment. We were unable to verify the expression of HSPG2 in the father as we did not propose muscle biopsy to the father, being an invasive procedure.

  1. In Figure 4, DAPI staining in control were faint. It is better to change to bright pictures.

We thank the reviewer for the suggestion. We have now modified the brightness of the pictures.

Reviewer 2 Report

In this paper the authors present a case report of a patient carrying two mutations in HSPG2 gene. One mutation is already been described and associated with Schwartz–Jampel Syndrome type 1 (SJS1) and the other is a novel mutation. Besides the major problem of formatting and presentation of the data, I'm wondering if the phenotype presented is due to the well know mutation better than the novel mutation; In my opinion the authors do not give any new insight to the pathology, or describing any correlation between the novel mutation and the phenotype observed.

The major issue that I found in this paper is the poor care in the formatting of text (i.e. different fonts fig.3; hyperlink not removed) in and images.

- The introduction is too repetitive and not easy to follow. At line 44 "SJS is allelic to Silverman-Handmaker type of dys-segmental dysplasia " what does it mean? 

- Poor quality of ALL figures. Figure 1 is not aligned. Where is the graph of the qPCR, why it has not been shown?

- Figure 4. You show a decrease amount in the IF of the Perlecan protein, can you quantify it? Fiji allow the quantification of the elements in the figures;

- Since you did not show any data concerning the qPCR results, would be worth it show at least a quantification of the protein. Did you try to do a WB to verify the protein level in the patient?

- Did you have any data about the parent? mRNA level/protein level...

Overall, I think that this paper needs major improvements, not all for the english but both concerning formatting and experiments. 

Author Response

Comments and Suggestions for Authors:

In this paper the authors present a case report of a patient carrying two mutations in HSPG2 gene. One mutation has already been described and associated with Schwartz–Jampel Syndrome type 1 (SJS1) and the other is a novel mutation. Besides the major problem of formatting and presentation of the data, I'm wondering if the phenotype presented is due to the well know mutation better than the novel mutation; In my opinion the authors do not give any new insight to the pathology or describing any correlation between the novel mutation and the phenotype observed.

Thanks for your comments. From a clinical point of view, the purpose of our work is to report a novel mutation and to review the clinical knowledge on this rare disorder to underline clinical hallmarks. From the molecular point of view, our aim is to verify if there are some mutations or a particularly mutated region of the gene HSPG2 gene.

The major issue that I found in this paper is the poor care in the formatting of text (i.e. different fonts fig.3; hyperlink not removed) in and images.

We double-checked the formatting of the text and especially of all the figures.

- The introduction is too repetitive and not easy to follow. At line 44 "SJS is allelic to Silverman-Handmaker type of dys-segmental dysplasia " what does it mean?

We modified the introduction to make it clearer and without repetitions.

We have replaced the reference 7 “Arikawa-Hirasawa, E.; Wilcox, W.R.; Le, A.H.; Silverman, N.; Govindraj, P.; Hassell, J.R.; Yamada, Y. Dyssegmental dysplasia, Silverman-Handmaker type, is caused by functional null mutations of the perlecan gene. Nat. Genet. 2001, 27, 431–434”, with reference “Stum, M., Davoine, C. S., Fontaine, B., Nicole, S. Schwartz-Jampel syndrome and perlecan deficiency. Acta Myol. 2005, 24, 89-92.

We preferred to remove the sentence from line 44, mentioned by the reviewer because it was somehow misleading and not essential.

- Poor quality of ALL figures. Figure 1 is not aligned. Where is the graph of the qPCR, why it has not been shown?

We have now improved the quality of all figures and Figure 1 was aligned

We reported our results in the text as the mean value of 2-ΔΔCt for both patients and controls. We have also included the graph of the qPCR analysis as required (Figure 4A end 4B).

- Figure 4. You show a decrease amount in the IF of the Perlecan protein, can you quantify it? Fiji allow the quantification of the elements in the figures;

We have shown a clear reduction in the intensity of the immunostaining; however, we thank the reviewer for the suggestion since the addition of quantitative data will surely strengthen our result.

In the manuscript we have added the following sentence “Further, we quantitatively analyzed perlecan protein by measuring fluorescent area on five randomly selected fields, both in control and patient samples. Our quantitative analysis of perlecan fluorescence, normalized by fibronectin fluorescent area, confirmed a significant minor amount of perlecan protein in patient muscle fibre (p < 0.01, Figure 4C).” We have now added Figure 4A and 4C.

- Since you did not show any data concerning the qPCR results, would be worth it show at least a quantification of the protein. Did you try to do a WB to verify the protein level in the patient?

We do not have muscle biopsy sample from the patient to perform a WB analysis, we used the tissue to perform RNA extraction and to prepare cryosections. However, we have shown our qPCR results as previously requested.

- Did you have any data about the parent? mRNA level/protein level...

In the parents we performed only the sanger sequencing from blood sample to verify the presence of the two variants detected in the proband. Unfortunately, we did not perform the muscle biopsy in the parents due to ethical considerations (considering it is an invasive procedure), hence we were unable to perform mRNA, real-time or immunostaining analyses to verify the mRNA level/protein level.

Reviewer 3 Report

I have reviewed a manuscript titled: "Novel HSPG2 gene mutation causing Schwartz Jampel syn-2 drome in a Moroccan family and review of the literature". There are some concerns that the authors would like to follow-up and further discuss.

These concerns are as follow:

1. In figure 2A, the pedigree diagram seems cannot represent the fact that the parents are of consanguineous relationship and also the fact that there were siblings of proband, thus, it seems there are still some room of improvement.

2. Furthermore, it seems that the sanger’s sequence of both parents were not found in the manuscript. It is essential to include the Sanger sequencing of the parents as it was claimed that each of the parents possessing one variant and transmitted to the proband.

3. It was wondered whether the other sibling(s) of the proband underwent any genetic testing? if yes, what is the results and is it possible to be included in this manuscript? If no, why no genetic testing was done on that subject? As the subject might be prone to be the carrier of the variant of the HSPG2 too. Please explain.

4. It was mentioned that one of variant (c.3888+1G>A) was classified as “likely pathogenic” according to ACMG guideline. Maybe I have overlooked, it was wondered what is the ACMG classification of the other variant of the proband? My search seems to result in a “variant of unknown significance”, I do not know whether it is true according to the authors work.

5. the authors also mentioned that there was another previous report that the physical manifestation has been self-resolved, it was wondered if the proband of that case also carry any “VUS” variants or two copy of  “pathogenic” variant according to ACMG guideline?

6. It was also wondered if the condition of the subject can be self -resolved or significantly improved, how to account for the disease manifestation or phenotype observed in SJS1? As it is like the proband become the “carrier” because he does not show any manifestation of the disease and given that the other variant is “variant of unknown significance” according to my search.

7. Is there any blood sodium or potassium test done on the proband or any family members like parents and siblings? If no, why blood potassium and/or sodium was not done for these myotonia cases?

8. For the qPCR results, Maybe I have overlooked again, I did not see any figure showing the results. This will be important to see that as the readers can also learn something more on the errors range of the decrease. I know the authors should have prepared that, can the authors show it?

9. Further, can the authors mention if the decrease in HSPG2 mRNA expression of any statistical significance?

10. It was strongly recommended to describe any variant mentioned in this manuscript according to the ACMG guideline at least once (if not the first time mentioned) in the manuscript for proper referencing to the variant.

11. For the review of SJS, it was wondered if there any indications of the type of SJS reported? SJS1A or SJS1B or SJS2?

12. For the review of SJS, it was wondered if there any figure (something like figure 2C or figure 1 of Stum et al 2006) summarizing the all variants reported in this mini-review so to better showing the likely “hot spot” as it was mentioned in the manuscript? 

13. For the review of SJS, I am very sorry to say that there are still a large room of improvement, for instance, it was mentioned that “Revision of previously reported cases doesn’t show clinical or outcome correlation with the different mutations, so correlation genotype-phenotype is not suggested (Table 1).” It is fine that the authors have such statement, but I will expect a good review will further elaborate on those. Nevertheless, to my disappointment I do not see anything further.

14. Another example, it was mentioned that “Several pathogenic mutations including splice site variants, resulting in whole exon skipping or creating premature termination codons, have been reported in the HSPG2 gene in association with SJS1. Most mutations are private and there is no evidence of a founder effect.”

Which mutations or variants did the word “several” referred to? Also, “Most mutations are private and there is no evidence of a founder effect.”

15. The sentence looks alike to “All but four mutations were private, and we found no evidence for a founder effect.” (Stum et al, 2006) Stum et al, did explain that in their main text while I did not see any discussion in this manuscript.

16. Given with the present form of discussion, I have huge reservation to say this is a “review” of literature, but it is more likely to be a summary of cases since 2000. (the cases before 2000 seems not included) (not ALL published cases as mentioned in the abstract were included) as I do not see any discussion or in-depth “review” of all included cases. It is okay to keep the present form, just do not use “review” in the title (maybe “summary of cases since 2000” etc.) to better reflect what is present in the main text.

17. I guess there were no any mentioning of the therapy or management within the main text while there is a sentence mention about that in the conclusion. I afraid maybe I am too old-schooled or suborned that I guess no new information shall be included in the conclusion. Maybe the authors can work on that by either discuss some more about treatment or clinical management in the main text.

18. Furthermore, it seems that the last few sentence of the conclusion and abstract looks extremely alike if not identical, maybe the authors would like to work further on those sentences.  

19. For the review of SJS, can the authors explain what is the Greek word “lamda” means in the first 3 rows of the table?

20. Line 162, the title “real-time” seems not completed, please amended. Thank you.

some grammatical mistakes, please have some Englishe editing. Thanks.

Author Response

Comments and Suggestions for Authors:

I have reviewed a manuscript titled: "Novel HSPG2 gene mutation causing Schwartz Jampel syn-2 drome in a Moroccan family and review of the literature". There are some concerns that the authors would like to follow-up and further discuss.

These concerns are as follow:

  1. In figure 2A, the pedigree diagram seems cannot represent the fact that the parents are of consanguineous relationship and also the fact that there were siblings of proband, thus, it seems there are still some room of improvement.

We have now improved the figure 2A adding to its legend the following sentence: “Distant consanguinity of the proband's parents (third cousins) was reported”.

The proband has only another brother (currently 16-year-old), without symptoms, hence due to ethical considerations, it was not possible to proceed with genetic analysis in his case.

  1. Furthermore, it seems that the sanger’s sequences of both parents were not found in the manuscript. It is essential to include the Sanger sequencing of the parents as it was claimed that each of the parents possessing one variant and transmitted to the proband.

As suggested by the reviewer, we modified the figure 2 and its legend as shown below:

Figure 2. (A) Pedigree of the family. Distant consanguinity of the proband's parents (third cousins) is reported.The proband (II1) was a compound heterozygote for mutations c.3888+1G>A and c.8464G>A in the HSPG2 gene. The asymptomatic parents (I1 and I2) were carriers for one mutation of the offspring. (B-C) Electropherogram of the two identified mutations in all family members tested and their localization on schematic representation of perlecan. Figure perlecan created with Biorender.com (Biorender, Toronto, Canada). (D) The amino acid alignment around the novel splice site mutation c.3888+1G>A was evolutionally conserved across a variety of different species.

  1. It was wondered whether the other sibling(s) of the proband underwent any genetic testing? if yes, what is the results and is it possible to be included in this manuscript? If no, why no genetic testing was done on that subject? As the subject might be prone to be the carrier of the variant of the HSPG2 too. Please explain

The proband has only another brother (currently 16-year-old), without symptoms, hence due to ethical considerations and restrictions, it was not possible to proceed with genetic analysis in his case.

  1. It was mentioned that one of variant (c.3888+1G>A) was classified as “likely pathogenic” according to ACMG guideline. Maybe I have overlooked, it was wondered what is the ACMG classification of the other variant of the proband? My search seems to result in a “variant of unknown significance”, I do not know whether it is true according to the authors work.

We confirm that the second mutation found in our patient c.8464G>A in exon 64 was classified as “uncertain significance” following the ACMG guideline. This mutation was already described by Arikawa-Hirasawa and colleagues [6] causing the skipping of exon and associate to another mutation, as in our case, causing loss of function of the perlecan. We added the classification “uncertain significance” for the c.8464G>A mutation in the manuscript.

  1. the authors also mentioned that there was another previous report that the physical manifestation has been self-resolved, it was wondered if the proband of that case also carry any “VUS” variants or two copy of  “pathogenic” variant according to ACMG guideline?

A previous report [Bauchè et al] with slight self-resolved symptoms , presented 2 variants “One was a C-to-T transition in exon 22 (c.2746C > T), predicted to replace an arginine residue by a premature stop codon (p.R916X), and to reduce the protein length by 79.1%. The second mutant allele was a G to A transition in intron 13, 15 base pairs downstream exon 13 (c.2826 + 15g > a). This intronic change was predicted to create a new acceptor splice site with a consensus value equal to 83.47 (+53.1%) and to strengthen a cryptic donor splice site with a consensus value equal to 96.67 (+5.3%).”

Following the ACMG guideline, the variant c.2746C > T is classified as “likely pathogenic”. It is not possible following the ACMG guideline for the variant c.2826+15G>A because the reference nucleotide at the 2826+15 is not “G” but “T”. Furthermore, they declare that the substitution G>A is located in intron 13, but the last base of exon 13 is 1654 and not 2826, as reported on LOVD (https://databases.lovd.nl/shared/refseq/HSPG2_NM_005529.5_table.html). However, the authors performed experiments on mRNA to evaluate its pathogenicity.

  1. It was also wondered if the condition of the subject can be self -resolved or significantly improved, how to account for the disease manifestation or phenotype observed in SJS1? As it is like the proband become the “carrier” because he does not show any manifestation of the disease and given that the other variant is “variant of unknown significance” according to my search.

We described only a slight functional improvement over the disease course, not a self-resolved condition. We think that it’s difficult that the proband may be considered a “carrier” because he still displays a full clinical phenotype including myotonia and skeletal features. On the other hand, anecdotal cases with spontaneous improvement have been described in the literature.

  1. Is there any blood sodium or potassium test done on the proband or any family members like parents and siblings? If no, why blood potassium and/or sodium was not done for these myotonia cases?

Thanks for your suggestion. In the proband blood electrolyte profile were always within normal limits. We did not report because pathomechanism of myotonia in SJS is not related to electrolyte levels; indeed, in previous papers is not reported. We agree it should be useful to underline this point and we add in paper.

  1. For the qPCR results, Maybe I have overlooked again, I did not see any figure showing the results. This will be important to see that as the readers can also learn something more on the errors range of the decrease. I know the authors should have prepared that, can the authors show it?

We thank the reviewer for the suggestion. We have now included a figure focusing on qPCR results (Figure 4A and 4C).

  1. Further, can the authors mention if the decrease in HSPG2 mRNA expression of any statistical significance?

The decreased expression in HSPG2 mRNA levels were statistically significant (p=0,0238). For statistical analysis we used Mann Whitney t-test to compare non-parametric data. Differences were considered significant at p < 0.05. The analyses were performed with GraphPad Prism version 5.0 (GraphPad Software, San Diego, CA). We have included these data in the manuscript.

  1. It was strongly recommended to describe any variant mentioned in this manuscript according to the ACMG guideline at least once (if not the first time mentioned) in the manuscript for proper referencing to the variant.

All the variants mentioned in the manuscript have been already described and published by other authors. For this reason, we consider off topic of our manuscript to analyse them according to the ACMG guidelines. Then, adding the description of any variant mentioned in the manuscript would make the table more difficult to read and too large to be displayed.

  1. For the review of SJS, it was wondered if there any indications of the type of SJS reported? SJS1A or SJS1B or SJS2?

We agree to your suggestion, but it is not possible to distinguish SJS1A and 1B just looking at literature data. Because in the cited articles, the subtype is not explicitly described, we decided to not report it. Although, SJS2 is clinically similar to type IB, we excluded all cases of SJS2, because this condition is caused by mutations in a different gene (LIFR gene on chromosome 5p13) which map to a different chromosome, hence, we considered it off topic.

  1. For the review of SJS, it was wondered if there any figure (something like figure 2C or figure 1 of Stum et al 2006) summarizing all variants reported in this mini review so to better showing the likely “hot spot” as it was mentioned in the manuscript?

We have now added figure 5, summarizing all the variants reported in this review to better show the probable "hot spots".

We have now added the following sentence in the results “Domains III and IV of the perlecan protein have the highest number of mutations identified so far in the HSPG2 gene, with 14 and 16 mutations, respectively (Figure 5), though domain V has identified the most frequent mutation c.11208-7G>A carried in 5 patients of different ethnicity (4 from Saudi Arabia and 1 from Tunisia) [10]. The other two more frequent mutations in the population were c.8464+4A>G and c.4741-10T>G identified both in 3 Tunisian patients (Table 1) [5,10]. Among 44 different mutations identified along the HSPG2 gene, the most common were 16 splice, 11 deletions and 10 missense (Table 1).

Legend of figure 5. Localization of all the variants summarized in this review on schematic representation of perlecan (the 2 mutations identified in our family are in bold).

  1. For the review of SJS, I am very sorry to say that there are still a large room of improvement, for instance, it was mentioned that “Revision of previously reported cases doesn’t show clinical or outcome correlation with the different mutations, so correlation genotype-phenotype is not suggested (Table 1).” It is fine that the authors have such statement, but I will expect a good review will further elaborate on those. Nevertheless, to my disappointment I do not see anything further.

A comprehensive evaluation of clinical findings of previous described cases has been summarized in the table 1.

We have now added the following sentence in the conclusion “Since domains III and IV of the perlecan protein have the highest number of mutations identified so far, they can be considered as a hot-spot region of the HSPG2 gene associated with the SJS1 phenotype (Figure 5). Several pathogenic mutations including splice site variants, resulting in whole exon skipping or creating premature termination codons, have been reported in this gene in association with SJS1. Most mutations are private and there is no evidence of a founder effect.”

  1. Another example, it was mentioned that “Several pathogenic mutations including splice site variants, resulting in whole exon skipping or creating premature termination codons, have been reported in the HSPG2 gene in association with SJS1. Most mutations are private and there is no evidence of a founder effect.”.Which mutations or variants did the word “several” referred to? Also, “Most mutations are private and there is no evidence of a founder effect.”

As described in figure 5 added, domains III and IV of the perlecan protein have the highest number of mutations identified so far in the HSPG2 gene. We may therefore conclude that there is a hot-spot region of the HSPG2 gene associated with the SJS1 phenotype. We have now added this concept to the "conclusion". In the results we have now added the most frequent mutation identified so far.

  1. The sentence looks alike to “All but four mutations were private, and we found no evidence for a founder effect.” (Stum et al, 2006) Stum et al, did explain that in their main text while I did not see any discussion in this manuscript.

In table1, we describe cases’ ethnicities to investigate a possible founder effect. However, even in populations where several cases have been reported (i.e., Tunisia or Turkey), different mutations were described, often involving different domains, in single families within the same ethnicity, so we concluded that there is no evidence for a founder effect. We have now added this sentence in the conclusion.

  1. Given with the present form of discussion, I have huge reservation to say this is a “review” of literature, but it is more likely to be a summary of cases since 2000. (the cases before 2000 seems not included) (not ALL published cases as mentioned in the abstract were included) as I do not see any discussion or in-depth “review” of all included cases. It is okay to keep the present form, just do not use “review” in the title (maybe “summary of cases since 2000” etc.) to better reflect what is present in the main text.

Thanks for your comment, we add a sentence in the text to clarify the purpose of our paper.

We reviewed all clinical, instrumental, bioptic and therapeutic data of genetically defined reported cases. There are other anecdotical cases with clinical but not molecular diagnosis, and so SJS is not confirmed in these patients. The molecular definition had been reported for the first time in the 2000 (Nicole S, Davoine CS, et al. Perlecan, the major proteoglycan of basement membranes, is altered in patients with Schwartz-Jampel syndrome (chondrodystrophic myotonia). Nat Genet. 2000 Dec;26(4):480-3. doi: 10.1038/82638. PMID: 11101850), this is the reason why we reviewed the reports since 2000.

To this point, we believe our work should be considered a literature review.

  1. I guess there were no any mentioning of the therapy or management within the main text while there is a sentence mention about that in the conclusion. I afraid maybe I am too old-schooled or suborned that I guess no new information shall be included in the conclusion. Maybe the authors can work on that by either discuss some more about treatment or clinical management in the main text.

We have now added treatment data in the table, also in the test and we add management data of the patients.

  1. Furthermore, it seems that the last few sentences of the conclusion and abstract looks extremely alike if not identical, maybe the authors would like to work further on those sentences

Thanks for your comment. We have now replaced the sentence “Literature review of published cases with molecular diagnosis confirmed that clinical picture- and not instrumental data- addresses to diagnosis, genotype-phenotype correlation was not possible. Therapy is mainly focused on symptom management; several drugs have been administered with different efficacy; we report the second case with spontaneous improvement.” in the conclusion with the sentence “A literature review of published genetically confirmed cases and our patient findings confirmed that the clinical picture - and not instrumental data- addresses to diagnosis.  However, the overlap of the patients' clinics regardless of the mutation identified and the heterogeneity of EMG results makes it not possible to identify a correlation between genotype and phenotype. Pharmacological and nonpharmacological therapies aim to reduce symptoms of myotonia. In literature cases, different therapeutic approaches are described with different efficacy (Table 1). We report the second case with spontaneous improvement.”

  1. For the review of SJS, can the authors explain what is the Greek word “lamda” means in the first 3 rows of the table? We have indicated the meaning of the symbol in the table legend

We have now added meaning to the lightness of the figure λ, “genetical aspects firstly described in Nicole et al, clinical aspects reported in Stum et al.”

  1. Line 162, the title “real-time” seems not completed, please amended. Thank you.

We corrected the title to “PCR analysis of HSPG2” as suggested.

Round 2

Reviewer 2 Report

1- qPCR analysis: Authors used 6 control and 1patient to evaluate the mRNA level of HSPG2. How you normalized the controls? Beside the SEM in the control is really high (0.88). How you run the statistic since you have only one patient? Is it no possible do statistic with only one sample.

2- Figure 4: The signal of the images is very low and also the quality. How did you perform the quantitative analysis?

Finally DOES NOT mean in the end. (immunostaining paragraph)

The paper need a revision on english, typo and deleted elements not really deleted ( the founded multiple times in the text)

Author Response

Comments and Suggestions for Authors

  • qPCR analysis: Authors used 6 control and 1patient to evaluate the mRNA level of HSPG2. How you normalized the controls? Beside the SEM in the control is really high (0.88). How you run the statistic since you have only one patient? Is it no possible do statistic with only one sample.

Thanks for your comments. We normalized mRNA level of HSPG2 by calculating the ΔCt values (Ct HSPG2 – Ct GAPDH) for all the controls and patient samples. We do have a ΔCt value of a single control sample that deviates from the other control samples. We have now excluded this value by performing a Grubbs’ Test to quantify outliers and modified the SEM accordingly.

For the patient result, we performed a statistical analysis using the triplicate of the PCR reaction. We are aware that this method is not correct. We have now modified the graph replacing the result of the triplicate reactions with the mean ΔCt value of the patient in order to show the result without the indication of a significant value. We have also modified the text replacing “a significant reduction” with “a decrease”.

2- Figure 4: The signal of the images is very low and also the quality. How did you perform the quantitative analysis?

The quality of Figure 4 is 300 dpi and the dimensions comply with the journal’s requirements. We have however improved Figure 4B and provided a brighter version.

The quantitative analysis was performed with ImageJ software, by measuring the perlecan fluorescent area normalized by fibronectin fluorescent area.

Comments on the Quality of English Language

Finally, DOES NOT mean in the end. (Immunostaining paragraph)

We have now replaced the word “finally” with “afterwards”.

The paper need a revision on english

We have now revised the English of the manuscript.

Reviewer 3 Report

Just 2 concerns:

1. for previous question 10, my bottom-line is: please use the ACMG guideline formation (e.g. :BRAF(NM_004333.6):c.1799T>G(p.Val600Gly) to show your 2 reported variants in the manuscript at least once, better the first time you use them in the main text. This is important to use a standardized way to describe the variant for better communication. If you can, please also do that for the literature review, if not, the reported variants are my bottomline. 

2. All the figures seems too low resolution to read and the microscope images are hard to see because of the low resolution and too dark. Please correct these problem.

Author Response

Comments and Suggestions for Authors

Just 2 concerns:

  1. for previous question 10, my bottom-line is: please use the ACMG guideline formation (e.g. :BRAF(NM_004333.6):c.1799T>G(p.Val600Gly) to show your 2 reported variants in the manuscript at least once, better the first time you use them in the main text. This is important to use a standardized way to describe the variant for better communication. If you can, please also do that for the literature review, if not, the reported variants are my bottomline. 

Thanks for your comments. We have now replaced in the manuscript the variant “c.3888+1G>A” with “HSPG2(NM_005529.7):c.3888+1G>A” and “c.8464G>A” with “HSPG2(NM_005529.7):c.8464G>A” following the ACMG guidelines. We have now replaced in Table 1, the title of column “cDNA mutation” with “cDNA mutation HSPG2(NM_005529.7)” and replaced all mutations following the ACMG guidelines. We have also now added “HSPG2(NM_005529.7):” to 3 mutations c.11208-7G>A, c.8464+4A>G and c.4741-10T>G described on page 19.

  1. All the figures seems too low resolution to read and the microscope images are hard to see because of the low resolution and too dark. Please correct these problem.

We have now improved the quality of all 5 figures increasing the resolution at 300 dpi. We have also modified Figure 2D.
